# The Efficacy of a Lower Dose of Everolimus in Patients with Advanced Neuroendocrine Tumors

**DOI:** 10.3390/cancers16223773

**Published:** 2024-11-08

**Authors:** Rodrigo G. Taboada, Angelo B. Brito, Ana Luiza Silva, Rui F. Weschenfelder, Rachel P. Riechelmann

**Affiliations:** 1Department of Clinical Oncology, A.C. Camargo Cancer Center, São Paulo 01509-010, Brazil; rodrigo.taboada@accamargo.org.br (R.G.T.); angelo.brito@accamargo.org.br (A.B.B.); 2Oncology Service, Hospital Moinhos de Vento, Porto Alegre 90560-030, Brazil; ana.mattos@hmv.org.br (A.L.S.); rui.fernando@hmv.org.br (R.F.W.)

**Keywords:** neuroendocrine neoplasms, neuroendocrine tumor, everolimus, toxicity

## Abstract

Everolimus works against certain types of advanced neuroendocrine tumors. However, the approved dose of 10 mg orally per day can be toxic, with nearly 25% of patients experiencing serious side effects. Preliminary studies have shown that lower doses of everolimus may also be effective, and clinical trials often allow for dose reductions. However, there have not been comparisons between the effectiveness of lower and higher doses of everolimus. Our retrospective study aimed to compare how long neuroendocrine tumors were controlled with lower doses versus higher doses of everolimus. We found no significant differences in how long patients were able to stay on treatment with either dose. This result supports the conduction of a randomized clinical trial.

## 1. Introduction

Everolimus is an oral inhibitor of the mammalian target of rapamycin (mTOR) with anti-cancer activity in multiple tumor types, including well-differentiated neuroendocrine neoplasms, named neuroendocrine tumors (NETs). Molecularly, NETs often depend on the activation of the mTOR pathway and angiogenesis [1,2,3]. Everolimus is an effective therapy in second or further lines for patients with gastrointestinal (GI), lung, or pancreatic NETs. Compared with a placebo, everolimus significantly prolonged median progression-free survival (PFS) times in patients with progressive advanced pancreatic NETs or GI/lung NETs in monotherapy and, by investigator assessment (but not by central evaluation), in combination with octreotide in advanced functioning GI or lung NETs [4,5,6].

Everolimus, at its approved dose of 10 mg once daily, can cause clinically relevant drug-related adverse events in approximately half of patients. Among the most common toxicities, of any degree, are stomatitis (65%) and skin rash (50%), and approximately one-third of patients experience infections, diarrhea, nausea, and fatigue [4,5,6]. Data from randomized clinical trials suggest that grade 3 or higher adverse events consist of stomatitis (9%), diarrhea (7%), infections (7%), anemia (6%), hyperglycemia (5%), and fatigue (3%) [4,5]. Studies of real-world patients have reported even higher rates of severe toxicities, with all-grade everolimus-induced pneumonitis occurring in nearly 20% of patients and grade 3–4 in about 8% of cases [7,8,9]. In a retrospective multicenter study from our group, 22% of patients had grade 3–4 infections, with 6% being opportunistic infections, and 3.6% of patients suffered related deaths [10]. This higher incidence of severe toxicity among our real-world patients likely reflects a more comorbid and fragile population and a longer follow-up for the onset of adverse events when compared with phase III trials (nearly double, from 17 to 33 months) [10].

In phase 1 clinical trials, everolimus at 5 mg orally per day efficiently inhibited the mTOR and offered better tolerance. A phase I study with everolimus evaluated oral doses of 5 and 10 mg/day and intravenous doses of 10 to 30 mg/week in patients with refractory solid tumors [11]. After evaluating the pharmacokinetic and pharmacodynamic effects (phosphorylation of S6 kinase, an efferent protein of the mTOR pathway) in peripheral mononuclear cells, the authors recommended oral doses starting from 5 mg/day or from 20 mg/week for future studies. Another phase I trial evaluated everolimus plasma levels and performed pre- and on-treatment tumor and skin biopsies to immunohistochemically analyze the expression of mTOR-related proteins with different dose schedules. Although a dose-dependent inhibition of the mTOR pathway was found, grade 3 limiting toxicity occurred in five patients due to stomatitis (10 mg/day cohort) and in four patients who received 70 mg/week intravenously. No grade 3 toxicity occurred with the dose of 5 mg/day [12].

Consistent with routine clinical practice, and anticipating toxicity, phase III trials usually anticipate two dose reductions from 10 mg to 5 mg per day and, subsequently, to 5 mg every other day. In addition, treatment interruption for a maximum of 4 weeks was allowed to manage adverse events. Even so, everolimus-related adverse events resulted in treatment discontinuation rates in phase III trials ranging from 12 to 24% [4,5,6].

Because the approved dose of everolimus of 10 mg daily leads to significant toxicity and everolimus at 5 mg daily efficiently inhibited the mTOR in phase I trials, we conducted a multicenter retrospective study to evaluate the efficacy of lower doses of everolimus in patients with advanced NETs.

## 2. Materials and Methods

Consecutive patients with an NET diagnosis and previous use of everolimus were identified retrospectively through medical records at A.C.Camargo Cancer Center (São Paulo, Brazil) and Hospital Moinhos de Vento (Porto Alegre, Brazil). The protocol received approval from the local ethics committee of each participant institution. Informed signed consent from individual patients was not required.

Eligible patients were 18 years or older with histologically confirmed locally advanced or metastatic unresectable NETs of gastroenteropancreatic, lung, or unknown origins; a well-differentiated histology of any grade according to the 2019 World Health Organization (WHO) classification; measurable disease according to the Response Evaluation Criteria in Solid Tumours (RECIST) v1.1; or radiologically progressive tumors, and each had received at least one dose of everolimus [13,14]. Patients with NETs from unknown primaries were included when the primary tumors were likely of gastroenteropancreatic origins based on immunohistochemistry, and providing a primary origin from gynecologic or urologic could be ruled out based on radiological investigations and immunohistochemical profile. Mixed neuroendocrine non-neuroendocrine neoplasms were excluded. Patients with carcinoid syndrome were allowed to have received somatostatin analogs concurrently.

Clinical data were collected from electronic medical charts. The best overall response to everolimus was defined as the best response recorded from the start until the end of everolimus use, as documented in medical charts.

The primary endpoint was time to treatment failure (TTF) in patients who received a mean daily dose of 7–10 mg (higher dose [HD]) or ≤6 mg (lower dose [LD]) of everolimus. Considering that many patients who start at 10 mg daily will require dose reductions, we stratified the dose groups according to the mean dose administered for each patient throughout the treatment duration—instead of categorizing patients based on the starting dose. The TTF was defined as the duration of everolimus treatment from the first day of the first cycle (C1D1) until tumor progression (as defined by the treating oncologist in medical charts), a treatment change for toxicity/intolerance, or death by any cause. The TTF was compared between the dose groups (LD vs. HD) and adjusted for prognostic variables (age at C1D1 of everolimus, tumor grade, and line of everolimus). Overall survival (OS), a secondary endpoint, was calculated from the C1D1 of everolimus to the date of death or last follow-up. Cox regression proportional hazard multivariable analyses for the TTF and OS were performed to adjust for the above prognostic variables.

Dose reductions were decided by the treating physician, and they could be upfront (for frailty or older age) or during treatment due to toxicities. The evaluation of toxicities associated with everolimus in real-world settings was not the aim of this study, as this was already reported by our group and others [7,8,10]. Somatostatin analogs are usually well-tolerated, with few adverse events [15].

Descriptive statistics were used to report means, medians (range), and frequencies. The Chi^2^ nonparametric test was used to compare categorical variables, and the Mann–Whitney U test was used to compare medians. The Kaplan–Meier method was used to estimate all time-to-event data, and differences in survival (TTF, OS) times were evaluated using the log-rank test. Median survival and median follow-up time were summarized. Multivariable analyses were conducted using Cox-regression models, which were used to adjust the dependent variables of the TTF and OS for the prognostic effects of age (continuous variable), tumor grade (1 and 2 vs. 3), and line of everolimus (1 or 2 vs. ≥3). Two-sided *p* values < 0.05 were considered significant. All statistical analyses were performed using the STATA IC/16.0 software (StataCorp, College Station, TX, USA).

## 3. Results

From August 2011 to September 2023, 92 patients were included: 74 (80%) received a mean daily dose of 10 mg (6.8–10) and were classified as the HD group, and 18 (20%) received a median dose of 5.1 mg (4.7–6) and were classified as the LD group. Because of toxicities, 12 (16%) patients in the HD group and 10 (55%) patients in the LD group required dose reductions during treatment (not upfront). Overall, eight (9%) patients started everolimus 5 mg because of older age and/or frailty, and among these, one patient required further reduction to 5 mg every other day.

Treatment discontinuation was observed in nineteen (25.7%) patients in the HD group: sixteen because of severe toxicities, one case due to logistic issues from the health care provider, one patient stopped everolimus because he wished to do so, and one stopped therapy after surgical intervention. Among the sixteen patients who discontinued treatment because of adverse events, five had grade 3/4 infections, four had grade 3 pneumonitis, one had grade 3 stomatitis, one had both grade 2 stomatitis and persistent grade 2 diarrhea, three had grade 3 skin rashes with one also presenting grade 2 diarrhea, one had severe hepatic steatosis with grade 3 transaminases elevation and grade 3 dyslipidemia, and one had grade 3 anemia/thrombocytopenia. Four (22.3%) patients in the LD group stopped everolimus because of toxicities: one because of grade 2 stomatitis and three because of grade 3 myelotoxicity. There were no toxicity-related deaths.

Both dose groups had similar characteristics at baseline, except for more grade 3 tumors and an older median age in the LD group (Table 1).

At a median follow-up time of 4.2 years, the median TTF was 9.2 months (Interquartile Range [IQR]: 3.7–32) for patients in the HD group and 7.2 months (IQR: 3.9–27) for those in the LD groups (log-rank *p* = 0.85; Figure 1). Excluding patients with G3 NETs (N = 17), the median TTF was 5.3 and 9.2 months in the LD and HD groups, respectively (log-rank; *p* = 0.92). The median ki67 index of the G3 group was 30%.

The TTF was not significantly different in the LD vs. HD groups (Hazard Ratio [HR]: 1.24, 95% Confidence Interval [CI]: 0.68–2.25; *p* = 0.47), even after adjusting for age at the C1D1 of everolimus (HR: 1.02; 95% CI: 1.01–1.04; *p* = 0.002), NET grade (grade 3 vs. 1/2; HR: 1.27, 95% CI: 0.95–1.71; *p* = 0.11), or treatment line (third or higher vs. first or second; HR: 1.55, 95% CI: 0.92–2.62; *p* = 0.09).

In the LD and HD groups, the median OS was 3.6 years (IQR: 1.4–6) and 6.5 years (IQR: 1.37–9.98; log-rank *p* = 0.57; Figure 2), respectively. In the Cox model, older age (HR: 1.03, 95% CI: 1.01–1.05; *p* = 0.007), grade 3 NETs (HR: 1.68, 95% CI: 1.15–2.47; *p* = 0.008), and everolimus in the third or higher lines (HR: 2.1; 95% CI: 1.05–4.13; *p* = 0.036), but not everolimus mean daily dose (HR: 1.32, 95% CI: 0.56–3.13; *p* = 0.53), were independently associated with OS.

## 4. Discussion

This multicenter retrospective study showed no significant differences in the TTF or OS according to an LD or HD of everolimus in patients with advanced and progressive NETs. This potentially offers advantages in terms of reduced toxicity and lower treatment-related costs.

Given the significant side effects associated with everolimus at 10 mg daily, including severe adverse events experienced by a notable proportion of patients in both clinical trials and real-world settings, the prospect of using a lower dose is particularly appealing. Reducing adverse events enhances patients’ quality of life and improves treatment adherence, as patients are often more willing to continue therapies that they tolerate well. Additionally, lower drug costs increase the accessibility of treatments, which is especially relevant when there are limited financial resources for health care, such as in developing and underdeveloped countries. Lower costs from lower doses of everolimus include the reduced cost of the drug itself (e.g., in Brazil, where everolimus’ price is linear with dosage) and the reduced costs associated with health care utilization (e.g., hospitalizations, laboratory/imaging tests to diagnose and monitor adverse events, and medications to treat them).

The ideal dose of everolimus for each patient remains to be defined, and it certainly involves pharmacokinetics parameters such as nutrition status and renal and hepatic functions. In that sense, pharmacokinetic studies could guide the appropriate dose levels to avoid undesirable adverse events while offering an effective dose. A prospective cohort study of Japanese patients with renal cell carcinoma observed that patients who developed adverse events had blood concentrations of everolimus after eight days of treatment that were twice as high as those who did not experience toxicities [16]. In a pivotal phase I clinical trial of everolimus in solid tumor patients, toxicities were more frequent in the 10 mg daily group when compared with patients who received 5mg daily, with thrombocytopenia, hyperglycemia, and diarrhea occurring only in the 10 mg cohort [11]. Associations between higher blood level concentrations of everolimus and toxicity have also been reported in studies with thyroid cancer, breast cancer, and mixed solid tumor patients, but not exclusively in patients with NETs. Patients with NETs likely represent a unique population in terms of pharmacokinetic parameters when compared with other tumor types, as they often present good renal function (unlike patients with renal cell carcinoma) and more commonly have extensive liver metastases (unlike hormone-positive breast cancer patients) [17,18]. Nevertheless, it is improbable that therapeutic drug monitoring will be investigated in patients with NETs, with NETS being a less common neoplasia and because everolimus is available as a generic drug worldwide. Therefore, studies such as ours are important for helping clinicians evaluate the appropriateness of lower doses of everolimus for NET patients.

Another relevant aspect is how lower doses of everolimus could negatively impact disease control. Our study did not find significant differences in the TTF or OS according to dose groups, although a numerical gain was observed for the HD group for both outcomes. Yet, such differences were lost when we adjusted the analyses for known prognostic factors such as age, tumor grade, and line of therapy. A recent retrospective single-center study that evaluated the efficacy and toxicity of everolimus in 52 NET patients also observed an inferior numerical, albeit nonsignificant, median progression-free survival (7.5 months versus 12.4 months) among patients who started at 5 mg daily versus those who started at 10 mg daily [19]. However, the authors did not adjust the progression-free survival for other prognostic factors, and it is not clear how the authors analyzed the patients who required dose reductions from 10 mg daily.

We included patients with G3 NETs because these patients had tumors that mostly behaved as G2 NETs. As we are aware that everolimus is approved only for G1 or G2 NET patients, we performed a sensitivity analysis of the TTF excluding G3 NET patients. Still, the TTF in the LD versus HD group was not significantly different when analyzing only patients with G1 or G2 NETs.

Low-dose everolimus has also been evaluated in breast cancer. In a retrospective study of 163 patients with HER2-negative hormone-positive breast cancer, everolimus combined with hormonotherapy led to a median PFS of 9 and 10 months for 10 mg/day and 5 mg/day doses, respectively [20].

The limitations of this study are primarily associated with its retrospective design, which inherently carries the risk of bias, including selection bias and confounding factors that may have influenced treatment outcomes. Markedly, the main limitation of our study lies in the reasons for patients to start everolimus at 5 mg daily: older age and frailty. Such observation explains the inferior OS of the LD group in the unadjusted analysis. In our LD group, 55% of patients who used a lower dose did so because they needed a dose reduction from 10 mg due to adverse events, and 45% used a lower dose upfront due to frailty/comorbidities. Therefore, the LD group inherently had a worse prognosis. We tried to compensate for this selection bias by adjusting the TTF and OS for age, treatment line, and NET grade. Yet, unmeasured variables could have impacted our results. Additionally, TTF, unlike progression-free survival, is not a standard endpoint to evaluate cancer-directed therapies for solid tumors. But, we chose TTF as our primary endpoint because it represents a combined measure of treatment effectiveness when it considers both disease control and patient ability to tolerate treatment; additionally, it is difficult to properly estimate progression-free survival in a retrospective study because of the variability of intervals between imaging tests and difficulties objectively measuring radiological tumor response and progression. Lastly, our sample size of 92 patients may have contributed to our inability to detect small but significant differences in outcomes between the HD and LD groups.

Based on the present findings, we think that older and frailer patients should start everolimus at 5 mg daily. For fit patients, clinicians should be more flexible with dose intensity, starting treatment with 10 mg daily but reducing this to 5 mg daily at the onset of grade 2 adverse events. In these cases, we do not think clinicians should maintain 10 mg daily until a grade 3 or 4 event occurs. Yet, our results are hypothesis-generating and set the rationale to investigate the antitumor effects of LD of everolimus in a controlled study. Indeed, we have launched the EVENET trial, a near-equivalence randomized phase II trial, which tests the efficacy of 10 mg daily versus 5 mg daily of everolimus in patients with advanced and pre-treated grade 1 or 2 gastroenteropancreatic or lung NETs (NCT06472388). In this trial, the primary endpoint is the progression-free survival rate at 12 months.

## 5. Conclusions

This study suggests that everolimus at 5 mg daily offers a similar efficacy to 10 mg daily in patients with advanced NETs, while it likely decreases toxicity and possibly decreases treatment-related costs. Our study provides insights into the prospective validation of a strategy that could effectively treat patients with NETs with reduced toxicity and lower costs and increase drug access to patients worldwide.

## Figures and Tables

**Figure 1 cancers-16-03773-f001:**
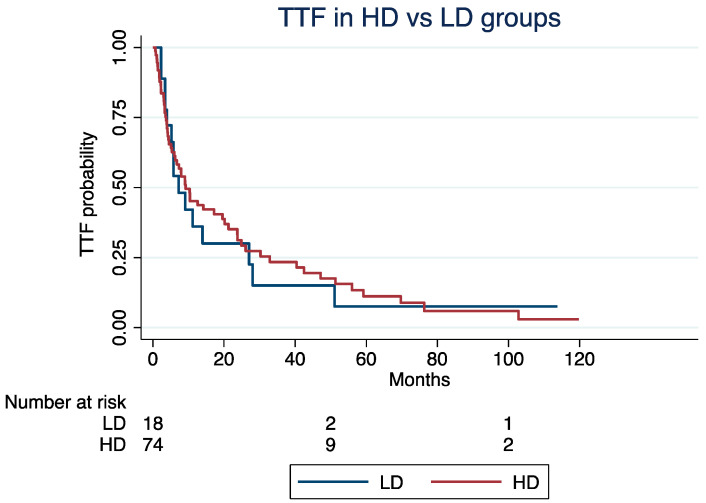
Time to treatment failure in high-dose vs. low-dose groups.

**Figure 2 cancers-16-03773-f002:**
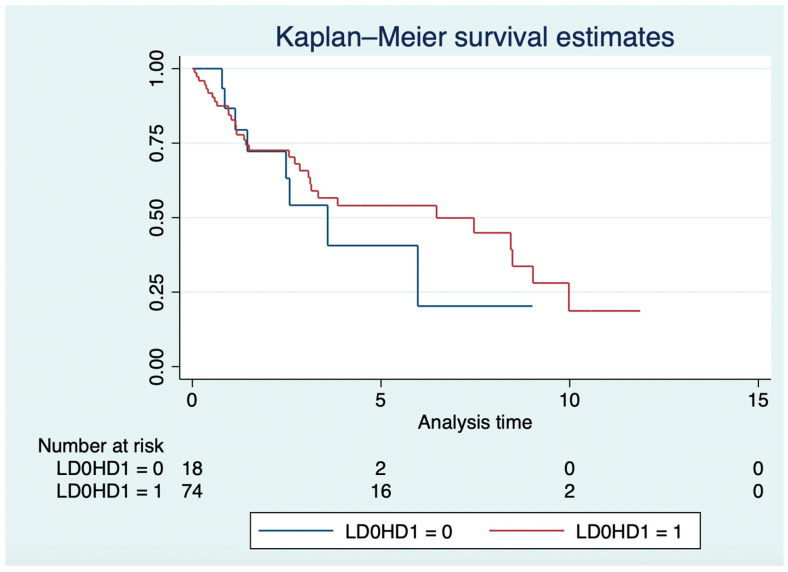
Overall survival in years of high-dose vs. low-dose groups.

**Table 1 cancers-16-03773-t001:** Baseline characteristics of high-dose and low-dose groups.

Characteristics	HD	LD	*p*-Value
Number	74 (100%)	18 (100%)	
Median age ^&^	53 (25–82)	68 (27–87)	0.007
Sex (% male)	47 (63%)	11 (61%)	0.8
ECOG *			0.1
0	42 (57%)	14 (78%)	
1/2	30 (41%)/2 (3%)	2 (11%)/2 (11%)	
Primary			0.8
Pancreatic	43 (58%)	10 (56%)	
GI/other	31 (42%)	8 (44%)	
Grade			0.003
1/2	9 (12%)/52 (70%)	3 (17%)/10 (56%)	
3	13 (18%)	5 (28%)	
Line of everolimus			0.46
1/2	12 (16%)/36 (49%)	3 (17%)/7 (39%)	
≥3	26 (35%)	8 (44%)	
Median dose (range)	10 (6.8–10)	5.1 (4.7–6)	-
Dose reductions for toxicity	12 (16%)	10 (56%)	-
Upfront reduction	-	8 (44%)	-

Abbreviations: ECOG, Eastern Cooperative Oncology Group; GI, gastrointestinal; HD, high dose; LD, low dose. ^&^ Mann–Whitney U test; all other comparisons, Chi^2^. * Percentages might not add up to 100 because of rounding.

## Data Availability

The data are available upon reasonable request.

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
