# Peer review of "The Efficacy of a Lower Dose of Everolimus in Patients with Advanced Neuroendocrine Tumors"

_cancers, 2024, doi:10.3390/cancers16223773_

Round 1

Reviewer 1 Report

Comments and Suggestions for Authors

Very interesting paper given that neuroendocrine neoplasms have been recorded in sharp increase in the last decade and collaterally the age of onset is decreasing, which was previously between the fifth and sixth decade of life. Unfortunately, even in pediatric age NETs are being found in patients hospitalized for appendicitis. In light of this, a point on everolimus must be made. Natale et al with a retrospective study have demonstrated on a population of about 100 people that lowering the dosage makes the drug more bearable with regard to adverse events with the same results. From the database , where there are complete data, it would be useful to link the efficacy of the drug with Ki67. They have read that at least in some cases everolimus was administered with somatostatin or analogues, some of the complications could therefore not be attributable to the former but to the latter (PMID: 38051513 to be cited in the bibliography). Furthermore, it is recalled that everolimus has the effect of inhibiting calcineurin and a few words should be spent on this too. In any case, this paper offers important ideas for research and reflection in good English, with discrete iconography and good bibliography.

Author Response

  • “Natale et al with a retrospective study have demonstrated on a population of about 100 people that lowering the dosage makes the drug more bearable with regard to adverse events with the same results.”

Response: Thank you for your positive comments on our study. Unfortunately, we are not aware of the Natale et al. study – could the reviewer forward its publication reference so we can include it? If this is a large series, it is relevant for us to mention.

Nevertheless, we cited the recent Kiesewetter et al. study, a single-center study with fewer patients and without adjusting for prognostic factors. Overall, the publication of similar studies (with similar results) in different populations within a short interval reinforces the subject's importance and points to the need for a randomized trial.

  • “From the database , where there are complete data, it would be useful to link the efficacy of the drug with Ki67.”

Response: It is valuable comment, thank you. Yet, because ki67 was obtained in distinct periods per patient, we do not trust this would be an accurate subgroup analysis. For example, some pts had the ki67 index measured in the primary NET, not in the metastatic lesion. We plan this analysis in the ongoing randomized trial, where patients will be stratified by ki67 index ( < 10% vs 10% or higher) upon randomization.

  • “They have read that at least in some cases everolimus was administered with somatostatin or analogues, some of the complications could therefore not be attributable to the former but to the latter (PMID: 38051513 to be cited in the bibliography).”

Response:  Thank you. We decided not to test the use of SSA in a subgroup analysis because:

  • Adding SSA to everolimus did not improve PFS in the COOPERATE-2 trial….
  • Somatostatin analogs are usually well-tolerated, with few adverse events. The evaluation of toxicities associated with everolimus, or somatostatin analogs, in real-world settings was not the aim of this study, as this was already reported by our group and others (PMID: 35641203, PMID: 25117065, PMID: 32403102). Nevertheless, we clarified this and cited the reference in line 125.

  • “Furthermore, it is recalled that everolimus has the effect of inhibiting calcineurin and a few words should be spent on this too”

Response: It is true, but the effect of everolimus on calcineurin is more important in the context of immunosuppression and solid organ transplantation and would not add, in our opinion, relevant knowledge to the context of our study.

Reviewer 2 Report

Comments and Suggestions for Authors

Interesting paper; however, there are some limitations First of all, the retrospective design is a major limitation to this study. The sample size is also quite limited.

Please add the number at risk to the KM curves.

Why did the authors use nonparametric tests ?

The discussion is quite short and should be improved. For example, the authors should comment on the general state-of-the-art in the field of advanced NETs, for example cite the recent study (PMID: 27956320)

Author Response

“Interesting paper; however, there are some limitations First of all, the retrospective design is a major limitation to this study. The sample size is also quite limited.

Please add the number at risk to the KM curves.

Why did the authors use nonparametric tests ?

The discussion is quite short and should be improved. For example, the authors should comment on the general state-of-the-art in the field of advanced NETs, for example cite the recent study (PMID: 27956320).”

  • “Please add the number at risk to the KM curves.”

Response: This has now been added to the Kaplan-Meier curves of TTF and OS by dose groups. Please see Figures 1 and 2.

  • “Why did the authors use nonparametric tests ?”

Response: The Chi2 nonparametric test was used to compare categorical variables (inherently discrete) since they do not require interval-level measurements. Mann-Whitney U test was used to compare medians and was chosen because of the non-normality distribution of the data due to the limited sample size.  

  • “The discussion is quite short and should be improved. For example, the authors should comment on the general state-of-the-art in the field of advanced NETs, for example cite the recent study (PMID: 27956320)”

Response: Thank you for your comments. However, we respectfully disagree that the discussion is short since it represents one-third of the manuscript. We understand your point about adding general information on NETs. But this is not the scope of this manuscript – readers can look for review articles on state of art management (as for example, our ASCO Education Book chapter in 2023 about Treatment Sequencing of NETs).

We are in doubt about citing the suggested reference “Primary tumour resection may improve survival in functional well-differentiated neuroendocrine tumours metastatic to the liver” because it is beyond the scope of this manuscript – we leave for Editors to decide if and where we should include this study Reference.

Reviewer 3 Report

Comments and Suggestions for Authors

Thank you very much for an interesting manuscript. 

1. One could imagine that the patients with the most co-morbidity startet Low dose Everolimus. Was it possible to adjust for co-morbidity?

Author Response

“Thank you very much for an interesting manuscript.”

  • “One could imagine that the patients with the most co-morbidity startet Low dose Everolimus. Was it possible to adjust for co-morbidity?”

Response: Thank you and we absolutely agree with you. Yet, a common limitation for retrospective studies with data extracted from medical charts is that comorbidity, but not always their severity, is documented. So, it would be inaccurate to adjust for comorbidity. Rather, we used objective prognostic factors, as such as age, line of treatment and tumor grade. We are collecting comorbidity information in our randomized trial EVENET.

Reviewer 4 Report

Comments and Suggestions for Authors

Dear Authors, 

this is an interesting multicenter and retrospective analysis on the clinical issue of the efficacy of 5 mg vs 10 mg everolimus in NET patients. 

I'd like to share these few observations about your manuscript, that I hope may improve your work. 

Minor issues: 

line 8: The authors can avoi to repeat “well-differentiated” that is already include in the definition of “NET”. Line 50: move the references before the fullstop.  Major Issues 96-97 lines: please specify that NET from “unknown primary” are likely from gastroenteropancreatic origin (for example, for the IHC or other features). Furthermore, everolimus is indicated only for G1 and G2 NET but the authors listed “any grade” in methods. I suggest to analyzed only patients with G1 and G2  GEP NET and typical or atypical carcinoid of the lung, according the guideline (ESMO 2020 ore ENETS 2024). 101-102 lines: The authors must to specified that only patients with carcinoid syndrome (and not other clinical  syndromes) were allowed to assume somatostatine analogs.

132-133 lines: I suggest the authors to remove G3 from the anaylisis because everolimus is not approved for these diseases. 

 156-157 lines: I suggest the authors to specify the grade of the toxicities also for LD group. 

197-198  lines: I suggest the authors to put more doubts about the results because the signifcance is not only the p in terms of OS (a difference of 3.6 ys vs 6.5 years is clinically significant and the fact that the authors had not a significant p is likely linked to the sample size...). 

Furhtermore, I suggest the authors, again, to remove G3 NET patients from the analysis of all the outcomes. 

Author Response

“Dear Authors,

this is an interesting multicenter and retrospective analysis on the clinical issue of the efficacy of 5 mg vs 10 mg everolimus in NET patients.

I'd like to share these few observations about your manuscript, that I hope may improve your work.

Minor issues:"

  • “line 8: The authors can avoi to repeat “well-differentiated” that is already include in the definition of “NET”.

Response: This has now been amended.

  • “Line 50: move the references before the fullstop.”

Response: This has now been amended.

  • “96-97 lines: please specify that NET from “unknown primary” are likely from gastroenteropancreatic origin (for example, for the IHC or other features).”

Response: Good point, thank you. This has now been added to the Methods section. Patients with NET from unknown primaries were included when likely of GEP, and providing a primary origin from gynecologic or urologic could be ruled out, based on radiological investigations and immunohistochemical profile.

  • “101-102 lines: The authors must to specified that only patients with carcinoid syndrome (and not other clinical syndromes) were allowed to assume somatostatine analogs.”

Response: This has now been amended.

  • “156-157 lines: I suggest the authors to specify the grade of the toxicities also for LD group.”

Response: Thank you, good point. This has now been amended.

  • “197-198 lines: I suggest the authors to put more doubts about the results because the signifcance is not only the p in terms of OS (a difference of 3.6 ys vs 6.5 years is clinically significant and the fact that the authors had not a significant p is likely linked to the sample size...).”

Response: We partially share the same point of view of the reviewer. The reviewer stated that the difference in the median overall survival was clinically significant, but when analyzing the Kaplan-Meier curves, it is possible to see that most of the events occurred early with both curves initially overlapping, reinforced in the Hazard-Ratio of 1.32 confidence interval of 0.56 – 3.13 and a p-value of 0.53. Additionally, the “clinical” difference was lost in the adjusted analyses for line of therapy, age and tumor grade, known prognostic factors for OS. Nevertheless, we agree that the small sample size could influence our results. This was stated as a limitation of our study:

“Lastly, our sample size of 92 patients may have contributed to our inability to detect small significant differences in outcomes between HD and LD groups.”

Taken together, these points reinforce our ongoing prospective randomized trial addressing this question.

  • “Furthermore, everolimus is indicated only for G1 and G2 NET but the authors listed “any grade” in methods. I suggest to analyzed only patients with G1 and G2 GEP NET and typical or atypical carcinoid of the lung, according the guideline (ESMO 2020 ore ENETS 2024).
  • 132-133 lines: I suggest the authors to remove G3 from the anaylisis because everolimus is not approved for these diseases.
  • Furhtermore, I suggest the authors, again, to remove G3 NET patients from the analysis of all the outcomes.”

Response:

  • We thank the reviewer for these comments. Although everolimus is not officially approved for patients with grade 3 NETs, it is often used in daily practice since some grade 3 NETs have more indolent course (“like G2-NET”). Thus, its inclusion reflects clinical practice (in specific settings) – the reason why these patients received everolimus in the first place. In fact, the median ki67 of the G3 group was 30%, thus, more of “low G3” NETs
  • Also, because many patients had their ki67 indexes evaluated in their primary tumors, the accuracy of ki67 as a marker of advanced NETs may not be perfectly accurate in daily practice.
  • Yet, after your suggestion and to better clarify the influence of grade, we performed a sensitivity analysis only with G1/G2 NETs patients and commented on this issue in the discussion section of the manuscript.

“Excluding patients with G3 NETs (N=17), the median TTF was 5.3 and 9.2 months in LD and HD groups, respectively (log rank; p = 0.92). The median ki67 index of the G3 group was 30%.”

Round 2

Reviewer 2 Report

Comments and Suggestions for Authors

I still think the paper has some limitations, such as the retrospective design and it is overall too short. True the discussion is 1/3 of the manuscript but it is 1/3 of a very short manuscript, so it is too short!

Author Response

Dear Reviewer, we are not sure how to address your concern. The manuscript contains the relevant information and we do not think it is necessary to write about the state of art of NET management. This would be beyond the scope of our study. 

regarding the study design, it is true that retrospective studies are limited. But when we evaluate the efficacy of less intensive regimens , it is unethical to do this in a prospective design without having retrospective data to support. Therefore, despite the limitations of our study, it is the best we can provide before embarking on a clinical trial with lower dose everolimus. Hope you are understanding of our arguments. 

Reviewer 4 Report

Comments and Suggestions for Authors

Dear authors, 

I think that you've improved the manuscript. 

Author Response

Thank you. We appreciate your positive feedback!